# Nitro Dihydrocapsaicin, a Non-Pungent Capsaicin Analogue, Inhibits Cellular Senescence of Lens Epithelial Cells via Upregulation of SIRT1

**DOI:** 10.3390/ijms232213960

**Published:** 2022-11-12

**Authors:** Pussadee Paensuwan, Thanet Laorob, Jatuporn Ngoenkam, Uthai Wichai, Sutatip Pongcharoen

**Affiliations:** 1Department of Optometry, Faculty of Allied Health Sciences, Naresuan University, Phitsanulok 65000, Thailand; 2Department of Chemistry, Faculty of Science, Naresuan University, Phitsanulok 65000, Thailand; 3Department of Microbiology and Parasitology, Faculty of Medical Science, Naresuan University, Phitsanulok 65000, Thailand; 4Department of Medicine, Faculty of Medicine, Naresuan University, Phitsanulok 65000, Thailand

**Keywords:** lens epithelial cell, diabetic cataract, senescence, oxidative stress, SIRT1

## Abstract

Diabetic cataracts are a common complication that can cause blindness among patients with diabetes mellitus. A novel nitro dihydrocapsaicin (NDHC), a capsaicin analog, was constructed to have a non-pungency effect. The objective of this research was to study the effect of NDHC on human lens epithelial (HLE) cells that lost function from hyperglycemia. HLE cells were pretreated with NDHC before an exposure to high glucose (HG) conditions. The results show that NDHC promoted a deacceleration of cellular senescence in HLE cells. This inhibition of cellular senescence was characterized by a delayed cell growth and lower production of reactive oxygen species (ROS) as well as decreased SA-β-galactosidase activity. Additionally, the expression of Sirt1 protein sharply increased, while the expression of p21 and phospho-p38 proteins decreased. These findings provide evidence that NDHC could exert a pharmacologically protective effect by inhibiting the senescence program of lens cells during diabetic cataracts.

## 1. Introduction

A diabetic cataract is a common complication in patients with diabetes mellitus leading to visual impairment and blindness [1,2]. The exact mechanisms of cataract development in diabetic individuals have been proposed in various metabolic abnormalities—one of these pieces of evidence being chronic hyperglycemic-induced oxidative stress [3]. For lens epithelial cells (HLE cells), glucose in the aqueous humor is the primary energy source and it sustains lens transparency [4]. Glucose enters HLE cells via glucose transporters (GLUT) [4]. Several studies have established that uncontrollable blood glucose levels contribute to lens oxidation and degeneration [5,6,7]. With aging, the oxidized lens proteins accumulate in lens tissue, causing lens opacities [5,8,9]. In addition, pathological features of the lens in diabetic cataracts are shown to be associated with cellular senescence [10,11].

Cellular senescence is a highly dynamic cellular process in cell fate regulation. Senescence is a momentous event that causes cellular aging involving an irreversible growth arrest state. It is prematurely induced by stress conditions, such as oxidative stress, DNA damage, and mitogenic stress [12,13]. Previous studies have demonstrated that long-term exposure to high levels of glucose triggers cellular phenotypic change promoting premature cell senescence [5,14,15]. Chronic hyperglycemia enhances the senescent program in lens cells, ultimately accelerating cataract-related diabetes [3,11]. Senescent cells disrupt metabolic homeostasis in lens tissues, creating a degenerative cycle of lens cells. Sirturin1 (Sirt1) is well-defined as a crucial regulator of cataract prevention. Previous studies showed a downregulation of the Sirt1 protein in high-glucose-promoted senescent cells [16,17]. Thus, targeting Sirt1 might prevent hyperglycemic-related cataracts.

Capsaicin (Cap), a trans-8-methyl-*N*-vanillyl-6-nonenamide, is a major pungent capsaicinoid in *Capsicum* spp., belonging to the family Solanaceae [18]. Cap is a group of acid amides condensing from vanillylamine and long-chain fatty acids [19]. The pharmacological effects of Cap comprise antioxidants, anticarcinogenic, anti-inflammation and neuroprotective effects [18]. Several lines of evidence reveal that the capsaicin-mediated activation of transient receptor protein vanilloid channel, TRPV1, could facilitate glycemic homeostasis and improve diabetes-related disorders. It has been shown that TRPV1 is activated as a mechanosensor, preventing cell shrinkage in response to hyperosmotic stimuli [20,21]. These suggest that TRPV1 is a potential therapeutic target in hyperglycemic-related lens diseases. Nevertheless, the functional role of TRPV1 in hyperglycemic-related lens disease remains unclear.

Much research has been conducted on the therapeutic effects of Cap in numerous medical application aspects [18,22,23]. However, the potential properties of Cap as a therapeutic alternative for ocular-related diseases are limited by its pungency effects and nociceptive activity [24,25]. Several attempts have been made to investigate Cap analogs to overcome the adverse effects of their pungency and burning-sensation properties. Recently, nitro dihydrocapsaicin (NDHC), a new synthetic non-pungent capsaicinoid, was constructed with a modification on the vanilloid residue of capsaicinoid derivatives [26]. NDHC was obtained by introducing the nitroarene group onto the aromatic ring in the Cap structure. By avoiding altering the hydroxyl group on vanilloid residues, NDHC retains a TRPV1 agonist property with a high binding affinity. Furthermore, the paramount properties of NDHC are its non-pungent effect and high stability.

Because prolonged high-glucose exposure is involved in the development of pathological lenses, the present study aimed to investigate the effect of NDHC, the new synthetic vanilloid derivatives, to alleviate HLE damage from hyperglycemic-induced senescence, which is related to Sirt1 upregulation. The beneficial effect of NDHC may lead to an alternative treatment for hyperglycemic-related lens disease.

## 2. Results and Discussion

### 2.1. Nitro Dihydrocapsaicin (NDHC) Synthesis

Regarding pharmacokinetics, Cap has a high bioavailability and bioactivity [18,22]. It is topically applied to treat various tissue pain syndromes [18,27]. However, Cap is highly pungent and can irritate the eye, causing ocular pain and a burning sensation [24]. Since the highly beneficial effects of Cap are limited by its pungency effect, the new NDHC was constructed to overcome this limitation with the aromatic ring modification of capsaicinoid derivatives, Cap (Figure 1A). The methoxy group replacement with the nitroarene group in the aromatic ring of Cap provided NDHC, a new Cap analog, with a low pungency effect (Figure 1B) [26,28].

### 2.2. Cell Viability Assay

Laorob et al. showed that the synthetic NDHC has a high-affinity binding to its cognate receptor, TRPV1 [26]. It has been shown that the replacement in the nitroarene group in NDHC enhanced the interaction between NDHC and TRPV1 by increasing π-stacking interactions [26]. This modification contributed to NDHC binding to the crucial amino acid residue of TRPV1: CDOCKER. Being a TRPV1 agonist, NDHC was studied for its effect on HLE cells. Western blot analysis revealed TRPV1 expression on HLE cells (Figure 2A). The level of TRPV1 expression did not significantly change in NDHC-treated cells compared with untreated cells (Figure 2B). Since TRPV1 is expressed on HLE cells; hence, NDHC might affect HLE cells through TRPV1 binding as previously stated [26].

To determine the lower pungency properties of NDHC, its cytotoxicity effect on HLE cells was examined. HLE cells were treated either with different concentrations of the NDHC or Cap. After 24 h treatment, there was no significant decrease in the viability of cells treated with NDHC at all concentrations. Nevertheless, the treatment with Cap was associated with a significant reduction in cell viability at the dose of 10 μM compared with NDHC (*p* < 0.01; Figure 2C). It has been shown that the methoxy group in an aromatic ring of Cap plays a major role in pungent induction [29]. It is possible that a pungency effect of Cap might be related to the irritation and toxicity to the cells. This is consistent with our results, in which HLE cells showed a decrease in survival following Cap treatment. Moreover, the potent cytotoxic effect was observed after Cap treatment in a dose-dependent manner (Figure 2C). These results agreed with a previous study, in which Cap at a a high dose could interfere with the function and permeability of the plasma membrane resulting in cellular damage [30]. Therefore, NDHC might have potential therapeutic application by overriding the pungent-limited Cap.

Since the pungency is very sensitive and depends on the molecular structure, the effect of prolonged exposure to NDHC in HLE cells was studied to determine the safe use of NDHC. The results show no significant impairment in the viability of NDHC-treated cells compared with untreated cells in a time-dependent manner (Figure 2D). These confirmed that the pungency effect of NDHC could be diminished by omitting the methoxy group in its aromatic ring. Collectively, our results confirm that NDHC synthesis by replacement of the methoxy group with the nitroarene group provided a cytoprotective effect, sustaining cell viability compared with Cap.

### 2.3. Long-Term HG Exposure Did Not Impair the Viability of HLE Cells

Based on the Expert Committee on the Diagnosis and Classification of Diabetes reports, the medium with 5.5 mM glucose (normal glucose, NG) was chosen to represent a normal blood glucose level. In contrast, the media with 35.5 and 55.5 mM glucose (HG) were used to describe the hyperglycemic blood according to the diabetic state defined by the oral glucose tolerance test (OGTT) with higher than 10 mM glucose level [7,31]. As the HLE cells could be damaged from high glucose levels, the effect of physiological change in glucose concentrations on cell viability was assessed.

The results showe that both HG conditions, 35.5 and 55.5 mM, did not significantly impair the viability of HLE cells in a time-dependent manner (Figure 3A). Next, we hypothesized that prolonged HG exposure may interfere with the growth of HLE cells. To explore this, the cell proliferation rate after exposure to 55.5 mM glucose for 5 days was focused (Figure 3B). Our results show that the occurrence of cell growth reduction was not significant after long-term exposure to HG compared to NG, which contradicts previous reports that HG can reduce cell viability and cell proliferation [7,31,32]. However, our results agree with several studies that high-glucose conditions did not influence cell viability [14,33,34]. One possible explanation is that exogenous glucose did not disturb survival-related signaling proteins [33]. We confirmed the sustainable viability of HG-treated cells by propidium iodide (PI) staining (Figure 3C). There were no observable PI-stained cells, indicating that cell membrane integrity was not damaged after exposure to HG. Under HG stress, it was found that there is an increase in myoinositol and choline in cell membrane, suggesting cytoprotective behavior of cells [35]. Together, a lack of a noticeable decrease in cell viability might suggest that HG did not influence cell proliferation. On the other hand, the viable HG-treated cells might have altered their metabolic activity to resist apoptosis and become senescent [12]. Thus, mimicking the hyperglycemic environment in vitro with 55.5 mM glucose was used in the subsequent experiments to study the HG-induced phenotypic changes in HLE cells.

### 2.4. NDHC Sustained the Viability of HLE by Alleviating the Oxidative Stress Response under High-Glucose Conditions

Next, the potential protective effect of NDHC from high-glucose-induced oxidative stress in HLE cells was studied. As shown in Figure 4A, pretreatment with NDHC did not significantly change the viability of HG-treated HLE cells in a time-dependent manner. However, HLE viability seemed to slightly increase in the NDHC-pretreated cells, suggesting that a prolonged exposure to NDHC was not cytotoxic. Furthermore, dramatic changes in cellular processes, including repair, cell death, and senescence, can be triggered in response to extrinsic stimuli. Senescent cells can be characterized in a hyperglycemic environment; in this case, cells remain viable with alterations in metabolic activity and phenotypes [12]. Therefore, NDHC pretreatment might prevent physiological cell changes and sustain the cell viability of HLE under HG conditions.

An imbalance between oxidants and antioxidants could generate cellular oxidative injury correlating with lens-related pathogenesis, including hyperglycemic-induced cataracts [15]. To study whether NDHC could modulate oxidative stress in HLE cells, the production of reactive oxygen species (ROS) was measured as a biomarker for the imbalanced redox potential. Upon NDHC pretreatment, HLE cells were cultured in a prolonged HG condition to induce oxidative stress. The fluorescent images clearly show that the HG condition increased the amount of ROS, confirming the oxidative induction of HG (Figure 4B). Interestingly, the excessive ROS level was inhibited by pretreating with NDHC (Figure 4B). The flow cytometric analysis of the ROS-positive cells probed by chloromethyl derivative of 2′,7′-dichlorodihydrofluorescein (CMH_2_DCFDA) clearly highlighted the increase in the number of ROS-positive cells during HG exposure compared to the NG condition (*p* < 0.05; Figure 4C). These results agree with previous studies, in which HG exposure induced oxidative stress [6,15]. In addition, the ability of NDHC to lower ROS increments under HG condition was significant (*p* < 0.01; Figure 4C). These findings indicate the potential antioxidative effect of NDHC in reducing ROS accumulation in HG exposure, thus enhancing cell viability and protecting cells from oxidative damage. Although the mechanisms underlying this antioxidation of NDHC are not known, the present results first described its sustainable antioxidant properties corresponding to its parent structure, Cap [36].

### 2.5. Enhancement of Cellular ROS Production and Senescence under HG Condition

Since robust cellular ROS production and the accumulation of senescent cells can be found in diabetic tissue [6,14], we hypothesized that HG conditions might facilitate the cellular senescence program through ROS production. To explore this notion, a cellular senescence-associated beta-galactosidase (SA-β-gal) activity was used as a senescence marker [37]. In particular, intense β-GAL labeling at pH 6.0 was found in senescent non-proliferating cells [38]. There was a significantly higher SA-β-gal activity in HG-treated cells compared to NG (*p* < 0.05; Figure 5A). In addition, 9H-(1,3-dichloro-9,9-dimethylacridin-2-one-7-yl) β-D-galactopyranoside (DDAOG) and p16-positive cells were assessed to confirm the accumulation of cellular senescence. DDAOG, a far-red fluorescent, was used as an SA-β-gal activity indicator following its cleavage by β-gal, providing different spectral properties [39]. The expression of p16^INK4a^ is also proven to be a senescence marker [40]. The results showed the marked accumulation of DDAOG and p16^INK4a^ positive cells in HG-treated cells (*p* < 0.05; Figure 5B,C). Although there was an insignificant decrease in cell viability, the increase in ROS production and SA-β-gal activity indicated that HLE cell senescence was a consequence of HG exposure. This might suggest that the overproduction of ROS was associated with high SA-β-gal activity, leading to cellular senescence.

As expected, we found that NDHC pretreatment could prevent HG-induced senescence of HLE cells by decreasing the expression of p16 coupled with the activation of SA-β-gal (Figure 5A–C). After NDHC pretreatment, we found a significant reduction in SA-β-gal activity in HG-treated cells corresponding with lower DDAOG expression levels compared to NG (Figure 5B). However, there was no significant decline in p16 expression in NHDC-pretreated cells compared with untreated cells. As p16^INK4a^ is a cell cycle regulator that restricts G1 to S-phase progression [40], it might indicate that advancing aged cells correlate with the obtained declining trends in HLE cell viability as preceding results. Together, these findings suggest a protective effect of NDHC on HG-related lens degeneration through its antioxidation and anti-senescence properties.

### 2.6. NDHC Reverses HLE Senescence by Upregulating SIRT1 and Inhibiting p21 and Phospho-p38 Expression

Since the process of lens cell degeneration in HG conditions may involve the function of various signaling proteins driving cellular aging outcomes, the influence of NDHC effect on downstream signaling proteins in HLE cells during HG treatment was investigated. The expressions of p21, phospho-p38 and Sirt1 in HG-treated HLE cells were evaluated in the presence or absence of NDHC. Western blot analysis revealed that the activation of p21 and phospho-p38 protein was significantly increased in HLE cells following HG treatment compared to nontreatment (*p* < 0.05, Figure 6A,B). These results agreed with previous studies, in which HG-induced diabetic cataracts involved the activation of p21 and p38 [31,41].

Sirt1 belongs to a family of histone deacetylases (HDACs). Notably, it has been reported that aging is related to the loss of Sirt1 expression, particularly in patients with senile cataracts [16,17]. A failure to deal with the HG of HLE cells might affect the anti-aging protein, Sirt1. The present results show a significant downregulation of Sirt1 in HG-treated cells compared to untreated cells (*p* < 0.05, Figure 6C). Interestingly, these protein expressions were significantly restored by pretreatment with the NDHC. This confirmed the potential effect of NDHC in preventing HG-induced degeneration through its antioxidative and anti-senescence effects. Our data confirm the potential effects of the application of NDHC in diabetic lens epithelial cell injury treatment. The supporting data are: (1) NDHC could diminish cellular ROS production and inhibit the senescence regulators, including SA-β-gal activity and p16 expression, in prolonged exposure to HG of HLE cells, (2) the NDHC counteracted aging-related intracellular proteins by abolishing p21 and phospho-p38 expression, and (3) NDHC induced the upregulation of Sirt1 expression, a well-known regulator in cataract prevention, withstanding prolonged high-glucose exposure.

## 3. Materials and Methods

### 3.1. Reagents and Antibodies

In this study, the following antibodies were used. Rabbit anti-p21, phospho-p38 and GAPDH antibodies were purchased from Cell Signaling (Danvers, MA, USA). Rabbit anti- Sirt1 antibody was from Abcam (Cambridge, MA, USA). Recombinant capsaicin was purchased from Abcam (Cambridge, MA, USA).

### 3.2. A Nitro Dihydrocapsaicin Preparation

In order to construct a nitro dihydrocapsaicin (NDHC), the process preparation of NDHC was defined as three steps, as previously described [26], following 3-nitro-4-hydroxylamine hydrochloride preparation as an aromatic amine salt, a coupling reaction between aromatic amine salt and 8-methylnonanoic acid, and purification. Briefly, aromatic aldehyde was converted into nitro-aromatic amine hydrochloride in 2 steps by nitration of aromatic aldehyde and followed by conversion corresponding aldehyde to primary amine group. Then, the NDHC was synthesized by coupling between primary amine and free fatty acid. Finally, the purification of the NDHC was performed by high- performance liquid chromatography.

### 3.3. Cell Culture

Human lens epithelial cells (HLE, clone B3; ATCC CRL-11421) were cultured in Minimum Essential Medium (MEM) (Gibco, Waltham, MA, USA) supplemented with 10% heat-inactivated fetal bovine serum (Gibco, Waltham, MA, USA) and 100 U/mL penicillin and 100 µg streptomycin (JRH Biosciences) at 37 °C in the humidified atmosphere with 5% CO_2_. Cells were subcultured with 0.05% (*v/v*) trypsin (Gibco, Waltham, MA, USA). Cells were plated overnight (24 h) before being used for experiments.

### 3.4. Cell Viability Assay

Cell viability was evaluated by the reduction in Alamar blue, from resorufin form to resazurin form, as an index of mitochondrial functional activity. Briefly, HLE cells were seeded into 96-well plates at a density of 5000 cells per well in complete growth medium. After 24 h of incubation, cells were treated either with 5.5 mM glucose or 55.5 mM glucose after a pretreatment with 10 µM of NDHC. After removing the medium with different conditions, 10% (*v/v*) of Alamar blue (ThermoFisher Scientific, Milano, Italy) was added into each well and incubated for 4 h at 37 °C. The relative fluorescence units of Alamar blue-containing medium were measured at 590 nm using a SpectraMax iD3 microplate reader (Molecular Devices, Sunnyvale, CA).

### 3.5. ROS Production

To assess intracellular ROS production, HLE cells were seeded into 96-well plates at a density of 5000 cells per well. Cells were probed with chloromethyl derivative of 2′,7′-dichlorodihydrofluorescein (CMH_2_DCFDA) for 20 min (ThermoFisher Scientific, Milano, Italy) according to the manufacturer’s instructions. Green fluorescence-stained cells were observed under a 20x objective lens of a fluorescence microscope (Nikon Eclipse Ti, Nikon^®^, Melville, NY, USA) using NIS-Elements D software. For some experiments, the CMH_2_DCFDA-probed positive cells were analyzed on a FACS Calibur flow cytometer (BD Biosciences, San Jose, CA, USA). Stained cells were analyzed for ROS production on a FACSCalibur (Becton Dickinson, Franklin Lakes, NJ, USA), and data were analyzed with CellQuestPro software.

### 3.6. SA-β gal Activity Assay

The senescence of β-galactosidase activity was detected based on the β-gal substrate 4-methylumbelliferyl β-D-galactopyranoside (4-MUG) (Cell Signaling, Danvers, MA, USA). After cell treatment, cellular lysates were collected for the determination of SA-β-gal activity according to the manufacturer’s instructions. Fluorescent intensity correlated with β-gal activation was measured at an excitation wavelength of 360 nm and an emission wavelength of 465 nm using a SpectraMax iD3 microplate reader (Molecular Devices, Sunnyvale, CA, USA).

### 3.7. Western Blot

For protein expression analysis, cells were pretreated with 10 µM of NDHC or left untreated and cultured in a medium with NG or HG at indicated time points. In order to collect HLE cells protein lysates, cells were then lysed in 100 µL Ripa buffer and phosphatase/protease inhibitor cocktail (ThermoFisher Scientific, Milano, Italy) for 15 min on ice. The cell lysates were subjected to SDS-PAGE and Western blotting with the desired antibodies, and visualization was carried out using a ChemiDoc XRS+ System (Biorad, Hercules, CA, USA). TheImageLab software version 6.1 (Biorad, Hercules, CA, USA) was used to assess the quantification of the band intensities.

### 3.8. Statistical Analysis 

Data are represented as mean ± standard deviation (SD). All differences between experimental groups were analyzed using Student’s *t*-test with Prism 9 software (GraphPad, La Jolla, CA, USA). Significant differences were considered when the *p* values were less than 0.05.

## 4. Conclusions

In summary, the present study first demonstrated that a synthetic nitro dihydrocapsaicin (NDHC), a novel capsaicin analog compound with non-pungent effects, exerts protective effects on lens cells under prolonged exposure to high glucose. The NDHC could be an alternative compound in therapeutic approaches for diabetic cataracts that decreased ROS production and abrogated cell senescence.

## Figures and Tables

**Figure 1 ijms-23-13960-f001:**
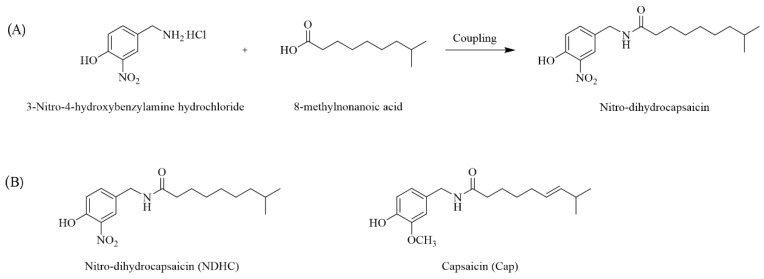
Schematic structure of nitro dihydrocapsaicin (NDHC). (**A**) Synthetic route of preparation of capsaicinoid analogs, NDHC. (**B**) Chemical structures of NDHC and capsaicin (Cap).

**Figure 2 ijms-23-13960-f002:**
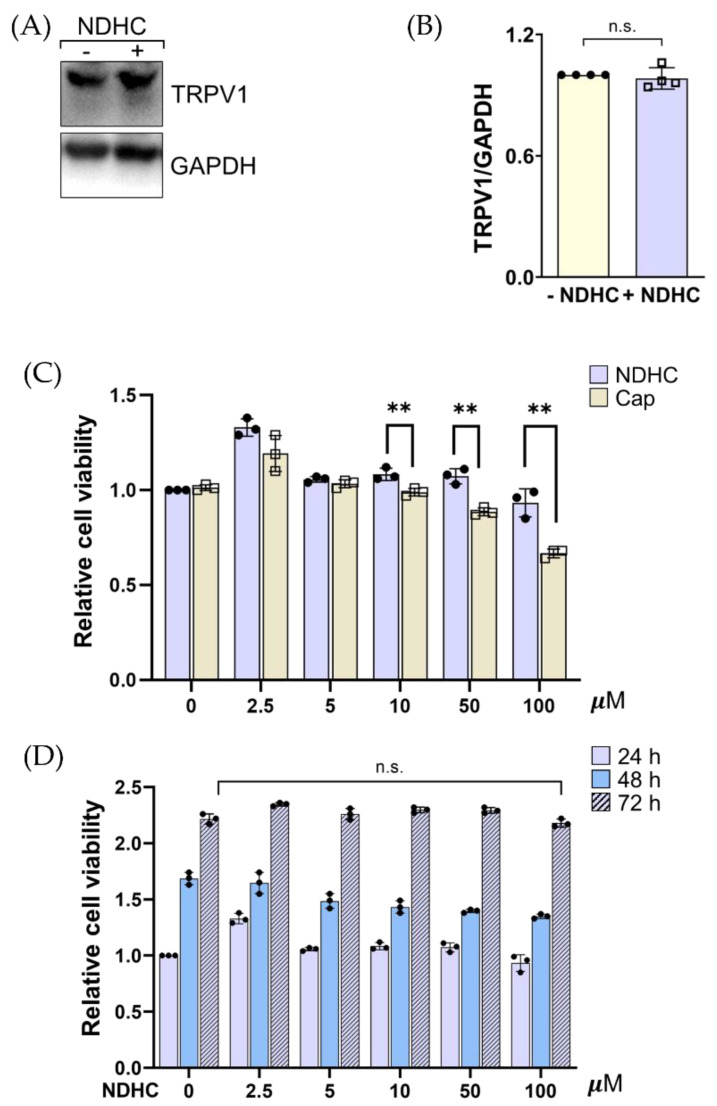
(**A**) Western blot analysis shows the expression of TRPV1 on HLE cells. After treatment with 10 μM NDHC, cell lysates were separated by SDS-PAGE and the Western blot was probed with anti-TRPV1 antibody. (**B**) Relative TRPV1 expression. Quantification of signal intensity, presented as the ratio of TRPV1 to corresponding GAPDH relative to the TRPV1 expression in untreated cells. (**C**) The viability of HLE cells treated either with NDHC or Cap at different concentrations for 24 h was examined by Alamar blue fluorescence (Ex 530, Em. 590 nm). (**D**) The viability of HLE cells treated with different concentrations of NDHC in a time-dependent manner was assessed by Alamar blue assay. Data are representative of 3 independent experiments (mean ± SD). ** *p* < 0.01, n.s. = not significant.

**Figure 3 ijms-23-13960-f003:**
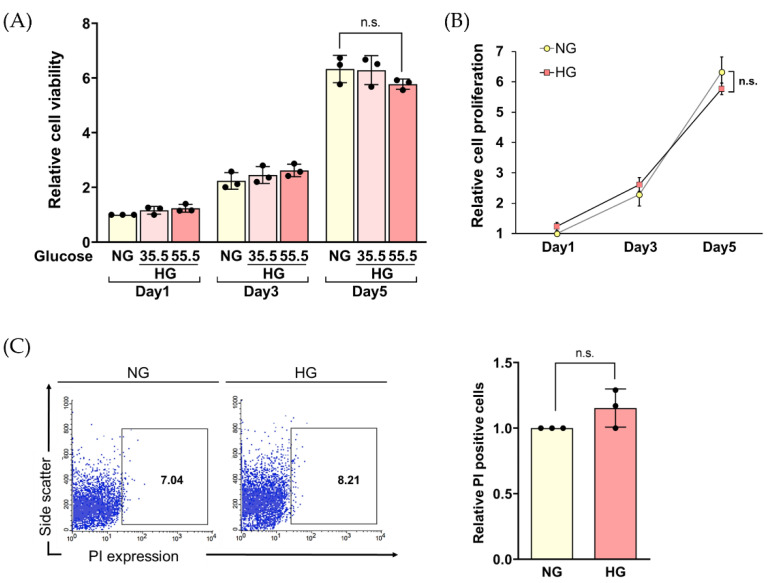
(**A**) The viability of HLE cells treated with changing glucose concentrations was examined for various lengths of time (Day 1, 3, 5) by Alamar blue assay. (**B**) Relative cell proliferation of HLE treated with 55.5 mM glucose at day 5 of incubation was determined by Alamar blue assay. (**C**) Flow cytometry was used to analyze the propidium iodide (PI)-stained cells after treatment with 55.5 mM glucose for 5 days. The relative PI-positive cell is shown in the graph, representing the mean ± SD. Data are representative of three independent experiments. n.s. = not significant, NG = normal glucose, HG = high glucose.

**Figure 4 ijms-23-13960-f004:**
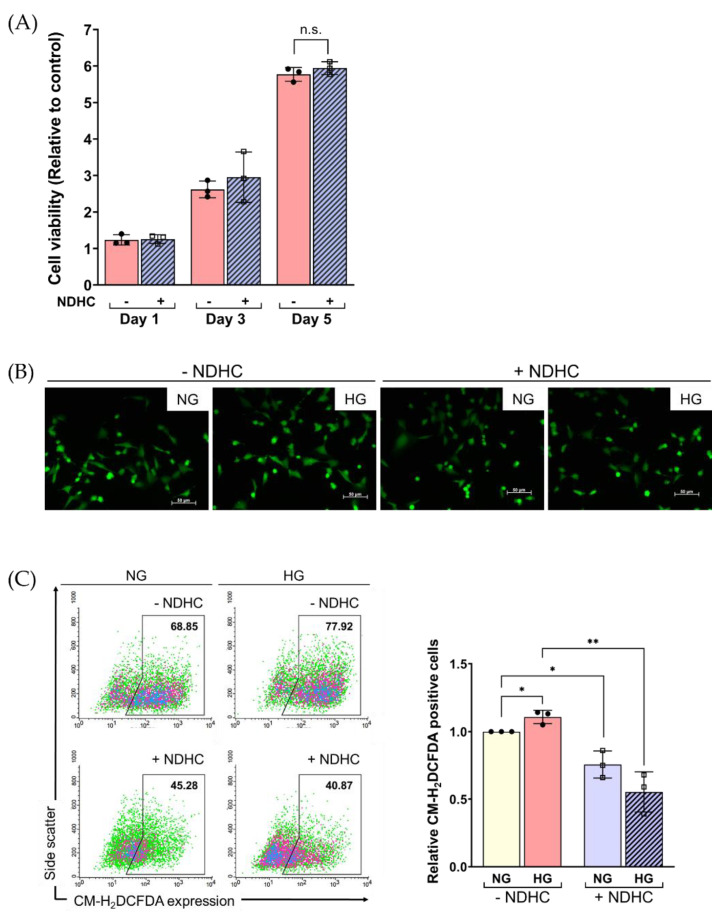
Effect of NDHC pretreatment on the viability of HLE and ROS production under high-glucose conditions. (**A**) The viability of HLE cell either pretreating with or without 10 µM of NDHC before exposure to 5.5 mM (NG) or 55.5 mM (HG) glucose for various lengths of time (Day 1, 3, 5) was evaluated by Alamar blue assay. (**B**) ROS expression was quantified by probing with chloromethyl derivative of 2′,7′-dichlorodihydrofluorescein (CMH_2_DCFDA). The green-stained cells were observed from a fluorescence microscope. (**C**) Quantitative analysis of the CMH_2_DCFDA-stained cell was obtained using flow cytometry. Data are representative of 3 independent experiments (mean ± SD). * *p* < 0.05, ** *p* < 0.01, n.s. = not significant.

**Figure 5 ijms-23-13960-f005:**
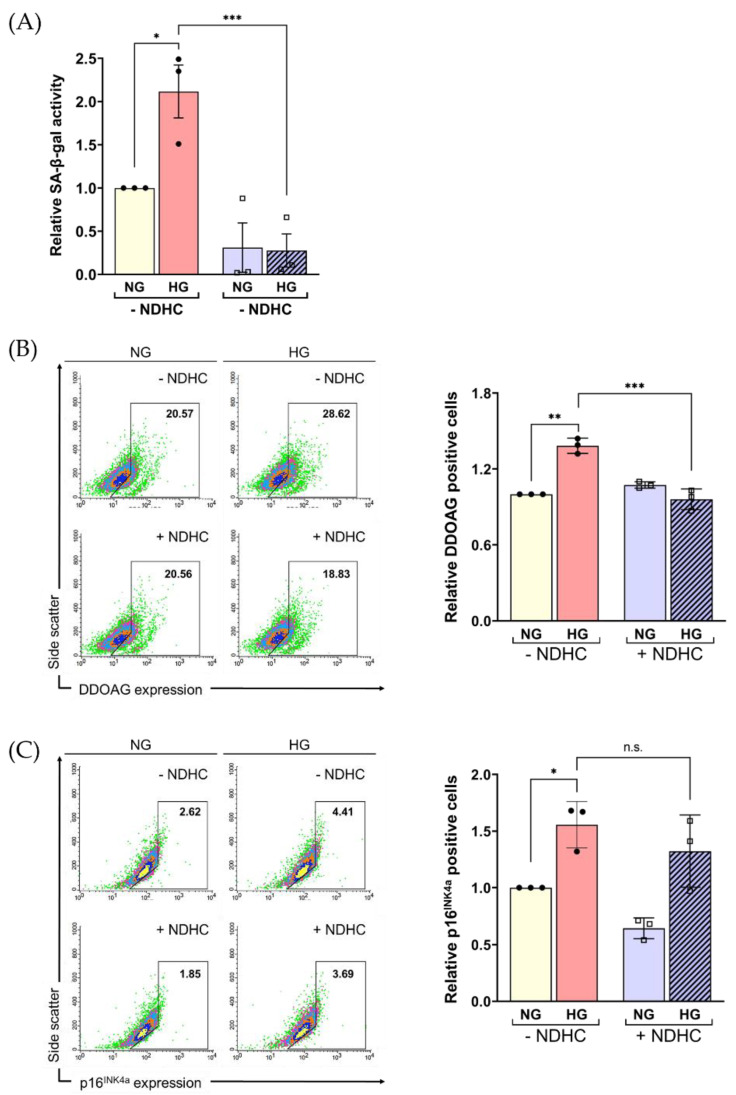
Effect of NDHC pretreatment on the senescence in HLE under high-glucose conditions. (**A**) Senescence-associated β-galactosidase (SA-β-gal) activity in HLE cells either pretreated with or without 10 µM of NDHC before exposure to 5.5 mM (NG) or 55.5 mM (HG) glucose was assessed for 5 days. (**B**,**C**) Flow cytometry was used to evaluate senescence-associated phenotypic changes. Cells were treated as in (**A**), and the percentage of DDAOG-positive and p16-positive cells was analyzed and is shown in the graphs as the mean ± SD. Data are representative of 3 independent experiments (mean ± SD). * *p* < 0.05, ** *p* < 0.01, *** *p* < 0.001, n.s. = not significant.

**Figure 6 ijms-23-13960-f006:**
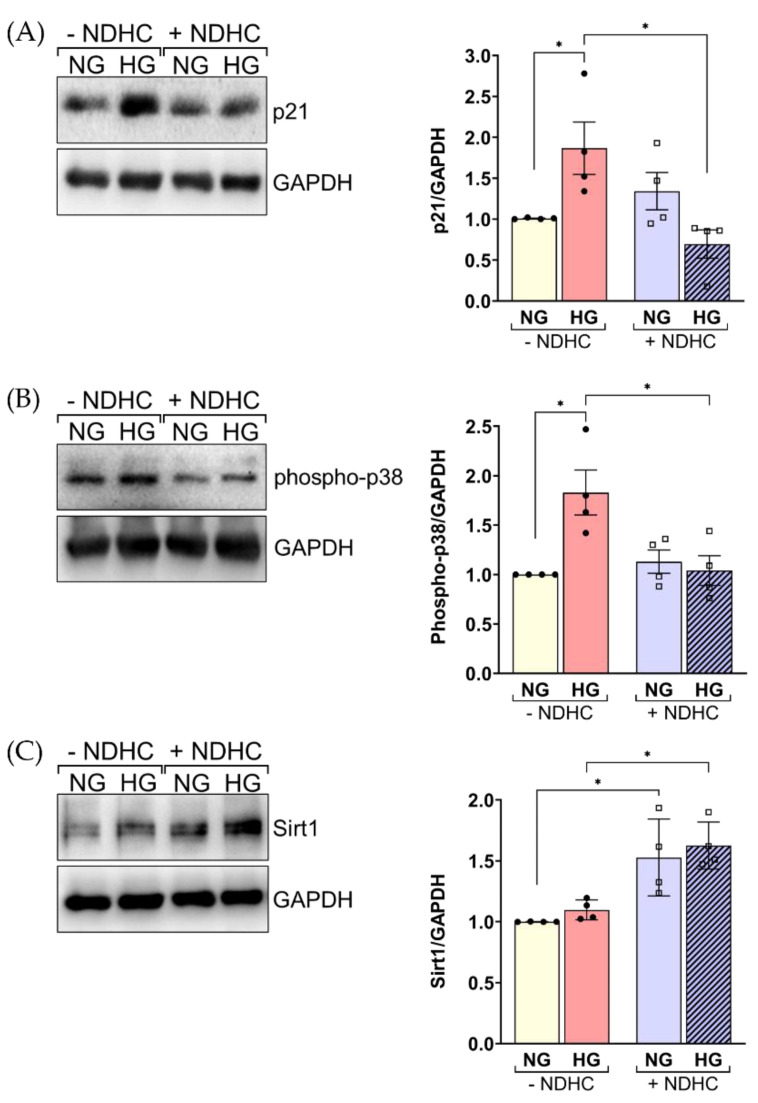
NDHD effects on senescence-related protein expression in HLE. HLE cells either pretreated with or without 10 µM of NDHC before exposure to 5.5 mM (NG) or 55.5 mM (HG) glucose for 5 days. Cell lysates were separated by SDS-PAGE and Western blot was probed with (**A**) anti-p21, (**B**) anti-phospho-p38 and (**C**) anti-Sirt1 antibodies. The quantified signal intensities are presented as a ratio of the protein of choice to the corresponding GAPDH values normalized to the value of unstimulated LEC. Data are representative of three experiments (mean ± SD). * *p* < 0.05.

## Data Availability

Not applicable.

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
