# Peer review of "Nitro Dihydrocapsaicin, a Non-Pungent Capsaicin Analogue, Inhibits Cellular Senescence of Lens Epithelial Cells via Upregulation of SIRT1"

_ijms, 2022, doi:10.3390/ijms232213960_

Round 1
Reviewer 1 Report
The paper is trying to investigate the effect of NDHC, a capsaicin analog, on human lens epithelial (HLE) cells in o high glucose culture conditions. Although the paper is well written in most parts, the experimental design is not very convincing enough to get the current conclusion. For example, the model used in the paper is not properly set up for experimental purposes. There is no discussion in this paper. The following problems must be properly answered and fixed.
1 In figure2, the author claimed they test the NDHC ligand binding in the HLE cells and made the statement that the present study demonstrated that HLE cells could be ligated with NDHC via its agonist ligand. However. Figure 1 does not provide any direct evidence showing that NDHC can bind with the TRPV1 receptor. Please use flow cytometry or other cell surface binding experiment (confocal imaging) to prove that NDHC can bind with TRPV1 on the cell surface.
Another question is why the author tested TRPV1 expression following NDHC treatment. What does this result tell us?
2 Figure2C, NDHC, and Cap group were significantly different following 10 uM treatment but not significantly different following 50 uM and 100uM treatment. Even though the data looks like the 50uM and 100uM groups had even bigger differences. Why is this? Can the author try to explain this data? What static comparison was used for these results?
3 All bar graphs in the paper need to be presented with individual data points.
4 There is no significant difference between groups. The author should not make this statement: However, there was a noticeable survival rate of HLE cells in 55.5 mM glucose-contained media that seemed to decrease following day 5 of incubation.” However, HLE viability seemed slightly increase in the NDHC-pretreated cells”. These are misleading.
5 As the author claimed and other studies showed that the HLE cells could be damaged by high glucose levels. However, the results in this work did not support this conclusion. The result lacks proper control to make such a conclusion. How does the author rule out the possibility of the test didn’t work? Does high glucose increase cell apoptosis? The author has to show evidence that the experiment setting is actually working.
6 Long-term HG exposure did not impair the viability of HLE cells. How did the author test the potential protective effect of NDHC in high glucose cultured HLE cells? This work did not set up a good model for the testing of the potential protective effect of NDHC.
Authors claimed that the potential protective effect of NDHC from high glucose-induced oxidative stress in HLE cells was studied. This is not right. The authors did not show that high glucose induces oxidative stress in HLE cells. This is just the hypothesis. The paper has not confirmed this yet. How can the author do further exponents based on one assumption? Can the author show if any oxidative stress proteins or signaling pathways are involved in this process? Data in figure 4B need to be quantified. In the non-NDHC treated group, the images of NG and HG were not taken in the same magnification. So, we can’t directly compare these two figures.
7 Figure6B, can the author use the total P38 to do the comparison?
8 There is no discussion for this manuscript.
Author Response
- In figure2, the author claimed they test the NDHC ligand binding in the HLE cells and made the statement that the present study demonstrated that HLE cells could be ligated with NDHC via its agonist ligand. However. Figure 1 does not provide any direct evidence showing that NDHC can bind with the TRPV1 receptor. Please use flow cytometry or other cell surface binding experiment (confocal imaging) to prove that NDHC can bind with TRPV1 on the cell surface.
Response: We thank the reviewer for this useful comment. Previous work has demonstrated that the synthetic NDHC has a high-affinity binding to its cognate receptor, TRPV1 [1]. The interaction results between the NDHC and TRPV1 have been evidenced using molecular docking experiments. The result has revealed that π-stacking interactions of NDHC and TRPV1 were enhanced by methoxy group replacement with the nitroarene group in the aromatic ring of Cap [1]. It has been shown that NDHC can bind to the crucial amino acid residues, CDOCKER, of TRPV1. These suggested that TRPV1 can be activated by binding with the NDHC through its crucial amino acid residues. Therefore, we hypothesized that NDHC might activate the HLE cells through its TRPV1. Then, we confirmed this hypothesis by determining the expression of the TRPV1 on HLE cells. We found TRPV1 expression on HLE cells, as shown in Figure 1. However, we agreed with the reviewer that we did not show the evidence of NDHC-TRPV1 interaction. Then, we have now rewritten a new sentence: "It has been shown that the replacement with nitroarene group on NDHC enhanced the interaction between NDHC and TRPV1 by increasing π-stacking interactions [26]. This modification contributed to NDHC binding to the crucial amino acid residues, CDOCKER, of TRPV1. Being a TRPV1 agonist, NDHC was studied for its effect on HLE cells. Western blot analysis revealed TRPV1 expression on HLE cells (Figure 2A). The level of TRPV1 expression did not significantly change in NDHC-treated cells compared with untreated cells (Figure 2B). Since TRPV1 is expressed on HLE cells, hence, NDHC might affect HLE cells through TRPV1 binding as previously stated [26], as highlighted on p.3, lines 113-120.
[1] Thanet Laolob, N.B., Neti Waranuch, Sutatip Pongcharoen, Wikorn Punyain, Sirirat Chancharunee, Krisada Sakchaisri, Jaturong Pratuangdejkul, Sumet Chongruchiroj, Filip Kielar, Uthai Wichai. Enhancement of Lipolysis in 3T3-L1 Adipocytes by Nitroarene Capsaicinoid Analogs. Natural Product Communications 2021.
Another question is why the author tested TRPV1 expression following NDHC treatment. What does this result tell us?
Response: This experiment was conducted to test for the presence of TRPV1 on HLE cells. In addition, we aimed to show that the inhibition of cellular senescence by NDHC was not due to the downregulation of TRPV1 expression but was caused by NDHC triggering TRPV1 that mediated the alteration of signaling pathway within HLE cells.
- Figure2C, NDHC, and Cap group were significantly different following 10 uM treatment but not significantly different following 50 uM and 100uM treatment. Even though the data looks like the 50uM and 100uM groups had even bigger differences. Why is this? Can the author try to explain this data? What static comparison was used for these results?
Response: We thank the reviewer of this value comment. We apologize for the missing for the statistic label on the graph. Now we have put the appropriate static label on the graph. We found that cell viability was significantly reduced from 10 uM of Cap onwards compared to NDHC. P values of less than 0.01 was summarized with two asterisks. We have added two significant asterisks for the 50uM and 100uM groups, as in Figure 2C. In the present study, we decided to use the concentration at 10 uM of NDHC at this concentration had no cytotoxic effect on HLE cells compared with Cap. For statical analysis, we use the student T-test with a p-value of less than 0.05.
- All bar graphs in the paper need to be presented with individual data points.
Response: We thank the reviewer for this informative comment. We have added two significant asterisks for the 50uM and 100uM groups, as in Figure 2C.
- There is no significant difference between groups. The author should not make this statement: However, there was a noticeable survival rate of HLE cells in 55.5 mM glucose-contained media that seemed to decrease following day 5 of incubation.” However, HLE viability seemed slightly increase in the NDHC-pretreated cells”. These are misleading.
Response: We thank the reviewer for this helpful comment. We have removed this confusing sentence out.
- As the author claimed and other studies showed that the HLE cells could be damaged by high glucose levels. However, the results in this work did not support this conclusion. The result lacks proper control to make such a conclusion. How does the author rule out the possibility of the test didn’t work? Does high glucose increase cell apoptosis? The author has to show evidence that the experiment setting is actually working.
Response: We thank the reviewer for this useful comment. Our study found that the viability of HLE cells was not disturbed after HG exposure which agrees with previous studies. Moreover, high glucose treatment did not induce cell apoptosis in this study, as we showed that PI-stained cells were increased even after five days of exposure to 55.5 mM glucose. These indicate that cell membrane integrity was not damaged after exposure to HG. Next, we examined whether prolonged HG exposure could induce cell senescence in HLE cells. To confirm this hypothesis, we found the occurrence of senescence phenotype in HG-treated cells. It correlated with the cell senescence definition that the senescent cells continuously slow their cell proliferation and cell entry to growth arrest and express the senescence phenotypes.
- Long-term HG exposure did not impair the viability of HLE cells. How did the author test the potential protective effect of NDHC in high glucose cultured HLE cells? This work did not set up a good model for the testing of the potential protective effect of NDHC. Authors claimed that the potential protective effect of NDHC from high glucose-induced oxidative stress in HLE cells was studied. This is not right. The authors did not show that high glucose induces oxidative stress in HLE cells. This is just the hypothesis. The paper has not confirmed this yet. How can the author do further exponents based on one assumption? Can the author show if any oxidative stress proteins or signaling pathways are involved in this process?
Response: In the current study, we aimed to investigate the anti-senescence effect of NDHC on long-term HG-treated-HLE cells through Sirt1 expression inhibiting. Since HG treatment might induce the senescence program through ROS production, the ROS contents were examined. We found that HG treatment induced the accumulation of ROS levels in HLE cells significantly, as shown in Figure 4B and 4C. We probed the ROS-positive cells with CMH2DCFDA and showed the green-positive cells using a fluorescence microscope (Figure4B). The number of ROS-positive cells was analyzed by Flow cytometry (Figure 4C). Moreover, Ros production was decreased in the NDHC-pretreated cells compared with the non-pretreatment (Figure 4B and 4C).
Data in figure 4B need to be quantified.
Response: We did not analyze the fluorescence intensity for the data in Figure 4B, but we analyzed the number of ROS-positive cells using flow cytometry as shown in figure4C. However, we used the same cell number, treatment conditions and dye for both assays.
In the non-NDHC treated group, the images of NG and HG were not taken in the same magnification. So, we cannot directly compare these two figures.
Response: We thank the reviewer for this useful comment. We confirmed that all pictures in figure 4B were taken under the same magnification. We have added the scale bar in all data in figure 4B.
- Figure6B, can the author use the total P38 to do the comparison?
Response: For figure6B, the band intensity of the proteins of interest was normalized with its corresponding GAPDH. All protein expressions were assessed from the same blots, and we used the same loading control, GAPDH, for normalization.
- There is no discussion for this manuscript.
Response: We wrote results and discussion as a combined section, under the section of “results and discussion”.

Reviewer 2 Report
The manuscript presents interesting data about using a non-pungent analogue of capsaicin in inhibiting/preventing the lens cell senescence in the conditions of diabetic cataracts. The study was generally well designed, however some minor changes in the text should be made as follows:
Results and discussion
Figure 2: The first sentence should be transferred to the Results section; please remove the point post ‘NDHC ligand binding’; the sentence ‘B) Relative fold change of TRPV1 …’ should be corrected.
Page 4: ‘(Figure 2C)’ should be transferred from the sentence ‘After 24-h treatment…’ to ‘Nevertheless, treatment …’
Figure 3: The first sentence should be transferred to the Results section; please explain, why ‘n.s.’ description was marked in the Fig. 3?; please separate ‘Day’ from ‘1,3,5’ in ‘(Day1,3,5)’
Figure 4: why ‘n.s.’ description was marked in the Fig. 4?; please separate ‘Day’ from ‘1,3,5’ in ‘(Day1,3,5)’
Page 7: please remove ‘full stop’ after significant before ‘(p<0.01; Figure 4C)’; please separate ‘5B from 5C in ‘(p<0.05; Figure 5B,5C)’
Figure 5: The sentence ‘B) Al treated cell were evaluated …’ should be corrected.
Page 8: Please change ‘were’ into ‘was’ in the sentence ’The expression of p21…’
Page 9: please separate ‘6A from 6B in ‘(p<0.05; Figure 6A,6B)’
Pages 9-10: Is the sentence ‘Moreover, the NDHC…’ finished?
Materials and Methods
3.3. Cell culture: Is ‘MEM’ abbreviation proper for ‘Minimum Essential Medium Eagle medium’?; how long the cells were cultured?; please add this information
3.6. Please write ‘western’ with a capital letter
Page 11: Please join ‘The’ with ‘ImageLab software version 6.1…’
Conclusions
Please change the font in ‘that decreased ROS production…’ (part of last sentence).
Author Response
Results and discussion
- Figure 2: The first sentence should be transferred to the Results section; please remove the point post ‘NDHC ligand binding’; the sentence ‘B) Relative fold change of TRPV1 …’ should be corrected.
Response: We thank the reviewer about this. We have removed the first sentence and the point post ‘NDHC ligand binding’. We have added a sentence; “After treatment with 10 ?M NDHC, cell lysates were separated by SDS-PAGE and the western blot was probed with anti-TRPV1 antibody” for Figure 2A, as highlighted on p.4, line 169-171. We have now rewritten a sentence in Figure1B to “Relative TRPV1 expression. Quantification of signal intensity, presented as the ratio of TRPV1 to corresponding GAPDH relative to the TRPV1 expression in untreated cells.”, as highlighted on p.4, line 171-173.
- Page 4: ‘(Figure 2C)’ should be transferred from the sentence ‘After 24-h treatment…’ to ‘Nevertheless, treatment …’
Response: We have now moved ‘(Figure 2C)’ to at the end of the sentence ‘Nevertheless, treatment …’ as highlighted on p.3, line 126.
- Figure 3: The first sentence should be transferred to the Results section; please explain, why ‘n.s.’ description was marked in the Fig. 3?; please separate ‘Day’ from ‘1,3,5’ in ‘(Day1,3,5)’
Response: We thank the reviewer for this. We have removed the first sentence. We added ‘n.s.’ to the data as shown in figure 3, following there was no significant cell viability and cell proliferation in HG-treated cells compared with NG-treated cells. We confirmed the viable HLE cells by PI staining. There were no observable PI-stained cells, indicating that cell membrane integrity was not damaged after exposure to HG. These results contradict previous reports in which HG can reduce cell viability and proliferation. However, several studies have shown that high glucose conditions might not always influence cell viability. We have discussed these points on p.5. We have separated ‘Day’ from ‘1,3,5’ as highlighted on p.5, line 229.
- Page 7: please remove ‘full stop’ after significant before ‘(p<0.01; Figure 4C)’; please separate ‘5B from 5C in ‘(p<0.05; Figure 5B,5C)’
Response: We have now deleted ‘full stop’ after significant before ‘(p<0.01; Figure 4C) and separated ‘5B from 5C as highlighted on p.7, line 311.
- Figure 5: The sentence ‘B) Al treated cell were evaluated …’ should be corrected.
Response: We have rewritten sentence in figure 5B to “Flow cytometry was used to evaluate senescence-associated phenotypic changes. Cells were treated as in A) and the percentage of DDAOG-positive and p16-positive cells were analyzed and shown in the graphs as the mean ± SD” as highlighted on p.8, line 383-386.
- Page 8: Please change ‘were’ into ‘was’ in the sentence ’The expression of p21…’
Response: We have now corrected ‘were’ to ‘was’, as highlighted on p.9, line 395.
- Page 9: please separate ‘6A from 6B in ‘(p<0.05; Figure 6A,6B)’
Response: We have separate ‘6A from 6B, as highlighted on p.9, line 398.
- Pages 9-10: Is the sentence ‘Moreover, the NDHC…’ finished?
Response: We thank the reviewer for this helpful comment. We have rewritten sentence to “and 3) the NDHC induced the upregulation of Sirt1 expression, a well-known regulator in cataract prevention, withstanding prolonged high glucose exposure.”, as highlighted on p.9, line 413-415.
Materials and Methods
- 3. Cell culture: Is ‘MEM’ abbreviation proper for ‘Minimum Essential Medium Eagle medium’?; how long the cells were cultured?; please add this information
Response: We apologize for the mistake at this point. We have corrected the full name of MEM to “Minimum Essential Medium (MEM)” and have added the detail for cell culture with a sentence “Cells were subcultured with 0.05% (v/v) trypsin (Gibco, Waltham, MA). Cells were plated overnight (24 h) before being used for experiments”, as highlighted on p.10, line 468-470.
- 6. Please write ‘western’ with a capital letter
Response: We have corrected ‘western’ to ‘Western’, as highlighted on p.11, line 503.
- Page 11: Please join ‘The’ with ‘ImageLab software version 6.1…’
Response: We have corrected ‘The ImageLab’ to ‘TheImageLab’, as highlighted on p.11, line 509-510.
Conclusions
- Please change the font in ‘that decreased ROS production…’ (part of last sentence).
Response: We have corrected the last sentence of conclusion part by font change, as highlighted on p.11, line 520-522.

Round 2
Reviewer 1 Report
Thank you so much for the revision. The authors answer most questions. however, the following two questions still need to be addressed.
1 All bar graphs in the paper need to be presented with individual data points.
Response: We thank the reviewer for this informative comment. We have added two significant asterisks for the 50uM and 100uM groups, as in Figure 2C.
Maybe the previous question is not clear. Instead of showing a white or black bar graph, all data points need to be shown on the bar graph, too. So the readers can see the distribution of the data points. It is hard to convince readers to show bar graphs without individual data points.
2 There is no discussion for this manuscript.
Response: We wrote results and discussion as a combined section, under the section of “results and discussion”.
A real discussion is needed for this manuscript. The current so-called results and discussion doesn't provide any real discussion. For example, can the author thoroughly discuss why some papers showed that the HLE cells could be damaged by high glucose levels and why some papers and this paper showed different conclusions? what is the biological significance of this work? et al.
Author Response
Responses to Reviewers
1. All bar graphs in the paper need to be presented with individual data points.
Response: We thank the reviewer for this helpful comment. We have now represented all graphs with individual data points, as shown in all figures.
2. There is no discussion for this manuscript.
Response: We thank the reviewer for this informative comment. We have now added a new sentence for more detail about the discussion, as shown below;
- “Nevertheless, treatment with Cap was associated with a significant reduction in cell viability at the dose of 10 ?M compared with NDHC (p<0.01; Figure 2C). It has been shown that the methoxy group in an aromatic ring of Cap plays a major role in pungent induction [29]. It is possible that a pungency effect of Cap might be related to irritation and toxicity to the cells. This is consistent with our results, in which HLE cells showed a decrease in survival following Cap treatment. Moreover, the potent cytotoxic effect was observed after Cap treatment in a dose-dependent manner (Figure 2C). These results agreed with a previous work, in which Cap at a high dose could interfere with the function and permeability of the plasma membrane resulting in cellular damage [30]. Therefore, NDHC might have potential therapeutic application by overriding the pungent-limited Cap.
Since the pungency is very sensitive and depends on the molecular structure, the effect of prolonged exposure to NDHC on HLE cells was studied to determine the safe use of NDHC. The result showed no significant impairment in the viability of NDHC-treated cells compared with untreated cells in a time-dependent manner (Figure 2D). These confirmed that the pungency effect of NDHC could be diminished by omitting the methoxy group in its aromatic ring. Collectively, our results confirmed that NDHC synthesis by replacement of the methoxy group with the nitroarene group provided the cytoprotective effect sustaining cell viability compared with Cap.”, as highlighted on p.3, lines 124-142.
- " Next, we hypothesized that prolonged HG exposure may interfere with the growth of HLE cells. To explore this, the cell proliferation rate after exposure to 55.5 mM glucose for 5 days was focused (Figure 3B). Our results showed that the occurrence of cell growth reduction was not significant after a long-term exposure to HG compared to NG, which contradicts previous reports that HG can reduce cell viability and cell proliferation [7,31,32]. However, our results agreed with several works that high glucose conditions did not influence cell viability [14,33,34]. One possible explanation is that exogenous glucose did not disturb survival-related signaling proteins [33]. We confirmed the sustainable viability of HG-treated cells by propidium iodide (PI) staining (Figure 3C). There were no observable PI-stained cells, indicating that cell membrane integrity was not damaged after exposure to HG. Under HG stress, it has been found that there is an increase of myoinositol and choline in cell membrane, suggesting cytoprotective behavior of cells [35]. Together, a lack of a noticeable decrease in cell viability might suggest that HG did not influence cell proliferation. On the other hand, the viable HG-treated cells might have altered their metabolic activity to resist apoptosis and become senescent [12]. Thus, mimicking the hyperglycemic environment in vitro with 55.5 mM glucose was used in the subsequent experiments to study the HG-induced phenotypic changes in HLE cells.”, as highlighted on p.5, lines 186-203.
- “Furthermore, dramatic changes in cellular processes, including repair, cell death, and senescence, can be triggered in response to extrinsic stimuli. Senescent cells can be characterized in a hyperglycemic environment, in this case, cells remain viable with alterations in metabolic activity and phenotypes [12]. Therefore, NDHC pretreatment might prevent physiological cell changes and sustain the cell viability of HLE under HG conditions.”, as highlighted on p.6, lines 242-247.
